# Sex difference in pathology of the ageing gut mediates the greater response of female lifespan to dietary restriction

Jennifer C Regan[1], Mobina Khericha[1], Adam J Dobson[1], Ekin Bolukbasi[1], Nattaphong Rattanavirotkul[1], Linda Partridge[1,2]*

[1]Institute of Healthy Ageing, Department of Genetics, Evolution, and Environment, University College London, London, United Kingdom; [2]Max Planck Institute for Biology of Ageing, Cologne, Germany

**Abstract** Women live on average longer than men but have greater levels of late-life morbidity. We have uncovered a substantial sex difference in the pathology of the aging gut in *Drosophila*. The intestinal epithelium of the aging female undergoes major deterioration, driven by intestinal stem cell (ISC) division, while lower ISC activity in males associates with delay or absence of pathology, and better barrier function, even at old ages. Males succumb to intestinal challenges to which females are resistant, associated with fewer proliferating ISCs, suggesting a trade-off between highly active repair mechanisms and late-life pathology in females. Dietary restriction reduces gut pathology in aging females, and extends female lifespan more than male. By genetic sex reversal of a specific gut region, we induced female-like aging pathologies in males, associated with decreased lifespan, but also with a greater increase in longevity in response to dietary restriction.

*For correspondence:
l.partridge@ucl.ac.uk

**Competing interests:** The authors declare that no competing interests exist.

## Introduction

Women live for longer than do men in most modern societies (*Regan and Partridge, 2013*) but suffer from higher levels of morbidity later in life (*Abad-Díez et al., 2014*; *Barnett et al., 2012*). Sex differences in health during aging are underpinned by differences in patterns of decline in the structure and function of specific tissues. For example, the gut ages differently in men and women, such that many gastrointestinal diseases and cancers are gender-biased (*Chang and Heitkemper, 2002*; *Kim et al., 2015*; *Jemal et al., 2011*). However, the mechanisms underlying sex differences in intestinal pathology are not well understood.

Females of the fruit fly *Drosophila melanogaster* show substantial gut pathology during aging, which limits female lifespan (*Biteau et al., 2010*; *Rera et al., 2013*; *Wang et al., 2014*). *Drosophila* are mostly post-mitotic as adults, but the gut contains intestinal stem cells (ISCs) (*Micchelli et al., 2006*; *Ohlstein et al., 2006*), and their division drives age-related intestinal hyperplasia (*Biteau et al., 2008*; *Choi et al., 2008*). Epithelial barrier function declines during aging, and its failure is predictive of death (*Rera et al., 2012*; *Clark et al., 2015*). However, it is not known to what extent males suffer from intestinal pathology during aging; indeed, studies of aging in male *Drosophila* are generally less common (*Magwere et al., 2004*; *Partridge et al., 1985*; *Tu et al., 2002*), and little is known about tissue-specific sexual dimorphisms in aging phenotypes (*Boyle et al., 2007*; *Camus et al., 2012*; *Mackenzie et al., 2011*). Interestingly, *Drosophila* females show a much greater longevity response to dietary restriction (DR) than do males (*Magwere et al., 2004*), although the role of the gut in this sex difference has not been investigated.

**eLife digest** Women live longer than men, and many age-related diseases are more common in one sex than the other. In addition, some treatments that extend the healthy lifespan of laboratory animals are more effective in females than in males. These include dietary restrictions, where food or specific dietary constituents are kept in short supply.

Stem cells can help to repair old and damaged tissue because, when they divide, they can form a cell that can specialize into one of several mature cell types. Previous studies of the fruit fly *Drosophila melanogaster* have shown that stem cell activity in the gut affects how long female flies live. Now, Regan et al. have looked in detail at the guts of male and female fruit flies as they age. This revealed that female guts deteriorate as the flies grow old because the stem cells in the gut divide more often and form small tumours. These stem cells help young females to grow and repair their guts, but start to turn against them as they age. In contrast, male guts stay well maintained and do not show the same signs of ageing.

Females fed less food had guts that aged more slowly, suggesting they might live longer on a restricted diet because it improves their gut health. Regan et al. then used a genetic trick to make male flies with female guts. These feminized males had more gut tumours than normal males, but they also showed a greater increase in lifespan when placed on a restricted diet, because the poorer condition of their ageing gut meant there was more scope for the diet to improve their health.

So if gut deterioration does not limit male lifespan, what do males die of? Pursuing this question may ultimately help us to understand how human lifespans are affected by sex differences and develop treatments for ageing and age-related diseases that everyone can benefit from.

We have uncovered substantial sexual dimorphism in the incidence of gut pathology during aging, with females showing widespread deterioration in epithelial structure and loss of gut barrier function, while males generally maintain both even at very late ages. However, males succumbed to oral bacterial infections to which females were resistant, with female guts containing a higher number of proliferating cells, suggesting a trade-off between gut plasticity and/or repair mechanisms, and old age pathology in females. Gut pathology in aging females was ameliorated by DR, suggesting that the greater response of female lifespan to DR may be a consequence of improved gut function. Taking advantage of cell autonomous sex determination in *Drosophila*, we produced males that had a region of the midgut genetically feminized and we found that this region alone showed the female pattern of aging-related pathology and increase in mitotic stem cells. Furthermore, these feminized males also showed a greater increase in lifespan in response to DR, comparable to that seen in females. Males demonstrated higher age-related, systemic inflammation compared to females, and sensitivity to oral bacterial infection, indicative of immune dysfunction, despite better maintenance of gut barrier function. Intriguingly, this points to further sex bias(es) in immunity that could contribute to male mortality.

## Results

### Sex differences in age-related gut pathologies

ISC division becomes dysregulated with age, leading to a hyperplastic intestinal pathology. ISC over-proliferation and a build-up of undifferentiated enteroblasts (EBs) lead to cell crowding and eventual tumor formation (*Biteau et al., 2008*; *Choi et al., 2008*; *Patel et al., 2015*). In order to better understand the effect of these events for intestinal epithelial organization, we subjected guts from outbred, wild type females and males with labeled epithelia ($w^{Dah}$;*Resille-GFP*) to high-resolution imaging over the entire adult lifespan at weekly intervals.

Several recent studies have shown the *Drosophila* gut to be highly regionalized in function, cell type and gene expression (*Buchon et al., 2013*; *Dutta et al., 2015*; *Veenstra et al., 2008*). We therefore analyzed four gut regions, identifying them with high fidelity (*Figure 1A*). These were the proventriculus (PV), midgut region 2 (R2) and midgut regions 4–5 (R4/R5), spanning most of the length and functional diversity of tissues in the adult midgut. Young females showed a well-

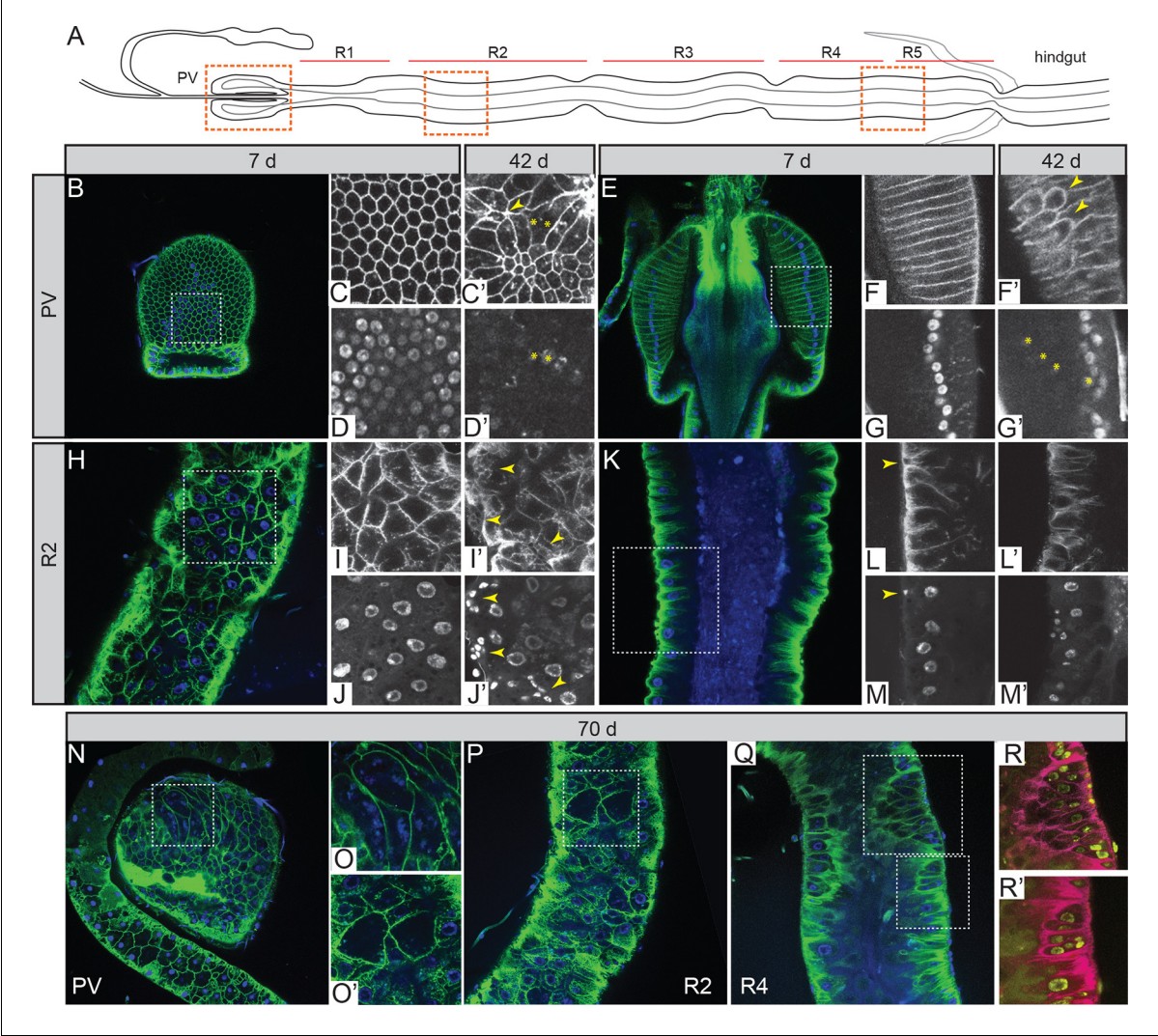

**Figure 1.** Intestinal stem cell activity produces severe epithelial pathology in females. (**A**) Outline of the adult gut indicating specific regions and areas subjected to image analysis (orange dashed boxes). (**B-D'**) Surface (**B**) and corresponding zoom (**C-D'**) of the proventriculus (PV) from 7-day (**B-D**) and 42-day (**C',D'**) –old females. Zoom panels show the green (epithelium; *Resille-GFP*) and blue (nuclei; DAPI) channels separately. Yellow arrowheads denote wound rosettes (**C'**) and yellow asterisks denote multinucleated cells (**C',D'**). (**E-G'**) Central section of the PV (**E**) and corresponding zoom panels (**F-G'**) in 7-day (**E-G**) and 42-day (**F',G'**) –old females. Yellow arrowheads denote extra, tumor-like cells in the epithelium (**I'**) and yellow asterisks denote their corresponding nuclei (**J'**). (**H-J'**) Surface of the gut at R2 (**H**) and zoom panels (**I-J'**), at 7days (**H-J**) and 42 days (**I',J'**). Yellow arrowheads denote small, tumor-like cell clusters. (**K-M'**) Luminal section at R2 (**K**) and zoom panels (**L-M'**). Yellow arrowhead denotes basal ISC (**L,M**). (**N-R'**) pathology in very old (70 day-old) females: PV surface (**N**) and corresponding zoom (**O**); R2 surface (**P**) and corresponding zoom (**O'**). (**Q-R'**) R4 section and corresponding zoom. Zoom panels (**R,R'**) have had their colors inverted to better visualize tumor nuclei.

The following figure supplement is available for figure 1:

**Figure supplement 1.** Epithelial pathology in females.

organized PV, with a honeycomb arrangement of tightly packed cells forming the outer wall (*Figure 1B–D*), arranged in a columnar epithelium with perfectly aligned nuclei (*Figure 1E–G*). In all distal midgut regions, large, polyploid, absorptive, enterocytes (ECs) were aligned to form a single layer epithelium with evenly spaced nuclei (*Figure 1H–M* and *Figure 1—figure supplement 1*). ISCs were nested at intervals along the basal side of the gut (*Figure 1K–M*) and were mitotically active as visualized by phosphohistone H3 (PH3) immunostaining (*Figure 2J*).

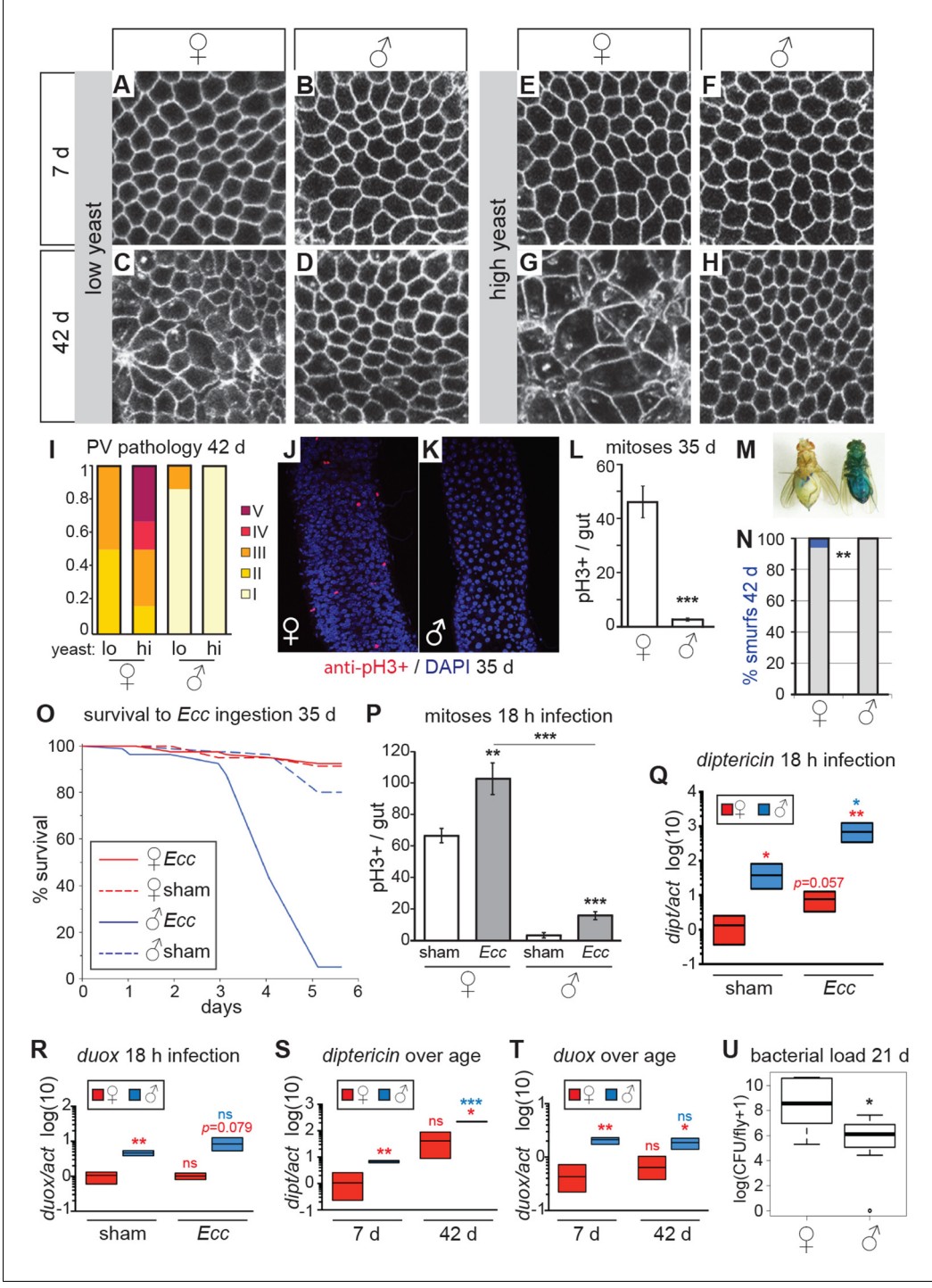

**Figure 2.** Females have more severe age-related intestinal pathologies than males. (A-D) Young (7 day-old) male and female flies had comparable epithelial organization in the PV (A,B), but at old age (42 days) only females showed epithelial pathology (C,D). For R2 region, see *Figure 2—figure supplement 1*. (E-H) Females raised on a high-yeast diet developed a more severe pathology than males by 42 days (G,H). (I) PV pathologies were binned into scaled categories, where I = WT undisrupted honeycomb, II = loss of regularity in epithelial cell size and pattern; few (<5) rounded unpolarized cells on apical side. III = sporadic wound healing rosettes and/or 5–10 apical cells; IV = widespread rosettes and/or >10 apical cells; V = severe pathology including holes, scars and tumors. Low yeast females tended to have less pathology than high yeast females (n=12 guts per condition, ordinal logistic regression, OLR; p=0.07) (J-L) Female flies had more actively dividing ISCs than males, visualized by anti-pH3+ immunostaining. Images from R5 in 35-day-old flies are presented (J,K); quantification of pH3+ cells

*Figure 2 continued on next page*

*Figure 2 continued*

per gut demonstrated that females had more mitoses than males at 35 days (n=20 guts per sex; student's *t* test, p < 0.001) (L). (M-N) Barrier function was compromised in old females but not males. 'Smurf' flies with leaky intestines (M) were present in female, but not male cohorts of *w*$^D$*;Resille* flies at 42 days (n≥150 flies per condition, representative of three repeated experiments, Fisher's exact, p = 0.008) (N). (O-P) Males succumbed to oral infection with the gram-negative bacterium *Erwinia carotovora (Ecc)* at 35 days, whereas females were resistant (O). PH3+ cell number per gut was increased in females (n≥10 per condition; student's *t* test, p = 0.0042) and males (p = 0.0003) upon *Ecc* oral infection. More mitoses were induced in female compared to male guts (p = 3.6E-05) (P). (Q-T) AMPs and ROS were higher in challenged and unchallenged males compared to females. *Diptericin* expression was higher in both sham- and *Ecc*-infected males, compared to females at 35 days (n≥3 samples per condition, 10 individuals pooled per sample, 2 technical repeats; *t* test with Welch's correction, p = 0.0135 for sham, p = 0.0012 for infected) and was upregulated significantly upon infection in males (p = 0.0132), and tended to be higher after infection in females (p = 0.0571) (Q). *Duox* expression was not upregulated after infection in 35-day-old males or females (n≥3 samples per condition, 10 individuals pooled per sample, 2 technical repeats; *t* test with Welch's correction, p = 0.8639 for females, p = 0.2303 for males), but was higher in males than females overall (p = 0.0060 for sham, p= 0.0793 for infected) (R). Systemic *diptericin* was higher in males than females and increased with age in males (n≥3 samples per condition, 10 individuals pooled per sample, 2 technical repeats; *t* test with Welch's correction, p = 0.0062 for 7-day-old females vs males; p = 0.0158 for 42-day-old females vs males; p = 0.2435 for 7-day-old vs 42-day-old females; p = 0.0003 for 7-day-old vs 42-day-old males) (S). *Duox* expression did not increase with aging in either sex, but expression was higher in males than females at both 7 and 42 days (n≥3 samples per condition, 10 individuals pooled per sample, 2 technical repeats; *t* test with Welch's correction, p = 0.0029 for 7-day-old females vs males; p = 0.0206 for 42-day-old females vs males; p = 0.4531 for 7-day-old vs 42-day-old females; p = 0.4857 for 7-day-old vs 42-day-old males (T). Males had a lower aerobic bacterial load than females at 21 days, (n≥8 samples per condition, 5 individuals pooled per sample; Wilcoxon test, p = 0.05) (U). A similar result was obtained for anaerobic load.

The following figure supplements are available for figure 2:

**Figure supplement 1.** Females have more severe age-related intestinal pathologies than males.

**Figure supplement 2.** Sex differences in pathology, barrier dysfunction and response to intestinal challenge.

In aging females, disruptions to the PV epithelium began to appear at between 2 and 3 weeks of age, with rounded, unpolarized cells clustering apically (*Figure 1F'-G'*). This abnormality was progressive (*Figure 1—figure supplement 1*), and eventually led to tumor formation. The effect of this hyperplasia on the basal side of the PV epithelium was marked; cells were pulled away from the basal edge due to tumor formation or apoptosis, leading to the appearance of rosettes characteristic of epithelial wound healing (*Figure 1C'* and *Figure 1—figure supplement 1*). These wounds became more numerous until, at very late ages, the epithelium did not heal properly, resulting in the formation of large scars, holes and multinucleated cells (*Figure 1C'-D', N-O* and *Figure 1—figure supplement 1*). R2 and R4 regions also showed dramatic aging phenotypes driven by an accumulation of small nuclei cells on the basal side - ISCs and EBs - which disrupted the single layer and eventually led to tumor formation and loss of epithelial organization (*Figure 1I',J',L',M',O'-R'* and *Figure 1—figure supplement 1*).

Strikingly, aged males showed only a low incidence of pathology in both the PV and midgut (*Figure 2A–H*, *Figure 2—figure supplement 1*). Indeed, many males in the oldest cohorts examined (up to 64 days post-eclosion) had well-maintained intestinal epithelia (*Figure 2—figure supplement 2A*). For example, outer wall cells of the PV were maintained as a single layer columnar epithelium with well-aligned nuclei (*Figure 2B,D,F,H*). The R2 midgut region retained regular spacing of ECs with few ISCs nested between them, similar to young guts (*Figure 2—figure supplement 1*). Aged males had far fewer PH3-positive cells in the midgut when compared to females (*Figure 2J–L*), suggesting that ISCs were largely quiescent. To determine whether this sex difference in ISC activity and epithelial integrity was reflected in barrier function, we fed aging *w*$^{Dah}$ females and males a blue dye that normally does not traverse the gut (*Rera et al., 2012*), at 42 days and 80 days, and scored flies that leached dye into the body cavity ('Smurfs'). A low, but consistent, proportion of Smurfs was found in old females, while males never produced Smurfs, even in the oldest (80 day-old) cohorts (*Figure 2M–N*, *Figure 2—figure supplement 2B*). This is striking given the maximum lifespan for

males (final surviving 10% of the population) is 64 days (*Figure 2—figure supplement 2D*). A recent study, describing the Smurf technique, found that the inbred laboratory strain $w^{1118}$ does produce male smurfs, but at a lower rate than females (*Rera et al., 2012*). We confirmed this, finding a small proportion of $w^{1118}$ male smurfs (<2%) at 42 days (*Figure 2—figure supplement 2C*).

## Males are more susceptible to intestinal infection and xenobiotic stress

While dysregulated ISC division may be detrimental at older ages (e.g. *Biteau et al., 2010*), ISC responsiveness to gut damage maintains intestinal homeostasis and promotes survival in young females (*Buchon et al., 2009*; *Chatterjee et al., 2009*).

We hypothesized that males, which have fewer ISCs and with a lower basal rate of division (*Jiang et al., 2009*; this study), would be more susceptible to intestinal stress such as oral infection, compared to females. The bacterium *Erwinia carotovora (Ecc)* induces antimicrobial peptide (AMP) expression and ISC division in the midgut when ingested (*Basset et al., 2000*; *Ayyaz et al., 2015*) but is not usually lethal to healthy adult females (*Ha et al., 2005*). When we challenged aged females and males with oral infection by *Ecc*, we found that females were resistant to the infection. However, a high proportion of males succumbed after around 72 hr (*Figure 2O*). Upon analysis of the ISC response to *Ecc* infection, we found that infected female midguts had a significantly higher number of actively dividing (PH3+) cells at 18-hr post-infection compared to controls. Although male midguts also responded to infection, the overall number of mitotic cells was significantly lower than in infected females (*Figure 2P*). In addition, when we exposed flies to a xenobiotic agent in their food (DDT; *Slack et al, 2011*), female flies were significantly more resistant to this challenge (*Figure 2—figure supplement 2E*). These findings suggest that, despite the patent hyperplastic pathology, older females are more resilient to intestinal challenge than are males. This sex bias may, in part, be promoted by higher numbers of dividing ISCs in females, despite this being detrimental to the maintenance of epithelial integrity at older ages.

Although repair of the epithelium is an important trait for survival during intestinal stress, other factors will also contribute. We therefore assessed the systemic response to *Ecc* oral infection in aged males and females. *Diptericin* (*dipt*, an antimicrobial peptide [AMP] responsive to gram-negative infection sensed by the Imd pathway), was expressed at higher levels in unchallenged males compared to females, and was strongly upregulated upon infection in males (*Figure 2Q*). This high systemic *dipt* did not induce a high survival rate for males, many of which later succumbed to the infection, and may instead be indicative of an infection not effectively contained by the gut and/or a sepsis-like response. To better understand inflammatory status during aging in females and males, we measured expression of *dipt* during aging. Only males showed increased expression of *dipt* as they aged and, at both ages, males expressed *dipt* at a significantly higher level than did females (*Figure 2S*). We also analyzed expression of the ROS-producer *dual oxidase* (*duox*) at young and old ages. *Duox* is expressed in immune-active epithelia, especially the gut, and is necessary for survival to foodborne pathogens (*Ha et al., 2005*). *Duox* levels did not change significantly during aging, but males had higher expression than did females at both ages (*Figure 2T*). This suggests that, even in the absence of acute infection, males suffer from a higher level of systemic inflammation than do females, despite better maintenance of gut barrier function.

The sex differences in gut pathology and systemic inflammation could affect the bacterial load in the intestinal microbiota, which could in turn affect pathology and inflammation. We therefore analysed internal bacterial load in females and males at 21 days, and found that females had a higher load than did males (*Figure 2U*). Higher bacterial load in the gut correlated with the higher levels of dysfunction and pathology observed in females compared to males (*Rera et al., 2012*; *Broderick et al., 2014*; *Clark et al., 2015*), and suggests that factors apart from total load, such as specific composition of the microbiota (*Broderick et al., 2014*; *Clark et al., 2015*), or infection through other routes (*Gendrin et al., 2009*), may explain the increased systemic *dipt* in males.

## Female intestinal pathologies are ameliorated by DR

ISC division is regulated by both diet and nutrient sensing IIS (*Choi et al., 2011*; *Biteau et al., 2010*; *O'Brien et al., 2011*). DR of dietary protein can extend lifespan in a wide range of animals (*Lee et al., 2008*; *Skorupa et al., 2008*; *Fontana and Partridge, 2015*; *Solon-Biet et al., 2015*). When we analyzed the guts of flies fed on two different yeast dilutions, we found that pathology

was lower in old females that had been exposed to the low-yeast diet (*Figure 2C,G,I*; *Figure 2—figure supplement 1*), in line with studies showing that ISC division (*Choi et al., 2011*; *O'Brien et al., 2011*) and intestinal barrier dysfunction (*Rera et al., 2012*) are reduced in females on restricted diets. Male flies, however, derived no obvious benefit to intestinal morphology from DR, largely because very little pathology was evident in males, even on a high-yeast diet (*Figure 2D,H,I*; *Figure 2—figure supplement 1*).

## Males with feminized midguts develop female-like intestinal pathology

Sex determination in fruit flies is cell autonomous, with X-chromosome counting leading to a cascade of splicing events and expression of sex-specific transcription factors (*Salz et al., 2011*). We exploited this cell autonomy, by mis-expressing the female-specific spliceform of *transformer* (*UAS-tra^F*) in male flies, using the midgut driver *NP1-Gal4* (*Zaidman-Rémy et al., 2006*), to achieve midgut-specific feminization. We then analyzed the effect on intestinal pathology during aging.

Cell size is larger in female flies than in males, in all tissues (*Alpatov et al., 1930*; *French et al., 1998*). Accordingly, males with feminized midguts had larger ECs compared to control males (*Figure 3—figure supplement 1A*). Quantitative PCR on dissected guts demonstrated that *doublesex* (*dsx^F*), the direct downstream target of *tra^F* (*Lee et al., 2002*), was expressed in feminized male guts at high levels (*Figure 3—figure supplement 1B*). As further proof-of-principle for sex reversal, we expressed *UAS-tra^F* in whole flies using the ubiquitous driver *da-Gal4*, producing transgendered males that were feminized in all sexually dimorphic structures (*Figure 3—figure supplement 1C*).

We analyzed intestinal aging in gut-feminized males (*w^D;NP1-Gal4/UAS-tra^F,Resille-GFP; NP1>tra^F*) and found that age-related pathologies were apparent at 35 days post-eclosion, whereas male control flies (*w^D;+/UAS-tra^F,Resille-GFP; +/tra^F*) generally had well-maintained guts at this age. Pathology in the feminized guts was comparable to that seen in females, or more severe (*Figure 3A–F"*). The *NP1-Gal4* driver is expressed throughout the midgut from R1, but only in a small number of cells in the PV (*Zaidman-Rémy et al., 2006*), serving as a useful internal control. Accordingly, the PV in *NP1>tra^F* males did not develop age-related pathologies (*Figure 3A–B"*, *Figure 3—figure supplement 2A*). PH3-positive cell number was dramatically increased in aged *NP1>tra^F* males compared to control males, as predicted by the observed pathologies (*Figure 3G*). When we analyzed barrier dysfunction in these flies, we found that at 42 days, gut-feminized males produced significantly more flies with a Smurf phenotype than control males (who produced none) and control females (*Figure 3H*). Feminized males presented a high level of systemic *dipt* expression (*Figure 3I*) and *Duox* expression (*Figure 3J*) at young and old ages. In addition, when we analyzed aerobic and anaerobic bacterial load at 7 and 21 days, we found that load increased with age as previously reported (e.g. *Broderick et al., 2014*; *Clark et al., 2015*). Importantly, while load in control females was higher than in males, this trend was switched in feminized males, who had a higher load than did females of the same genotype (*Figure 3K*).

To investigate whether midgut homeostasis in feminized males was responsive to diet, we raised *NP1>tra^F* flies on two different yeast dilutions. Similar to females, feminized males on the high-yeast diet appeared to show a more severe pathology at 35 days than did those on low yeast (*Figure 3L–M*). In line with this, when we analyzed the rate of tumor formation (category IV) in a specific region of the gut (R2), we found that DR significantly decreased the risk of tumor formation in feminized males (*Figure 2L*). The observed pathologies demonstrate that female and feminized ISCs respond to DR, however, when we quantified PH3+ cell number in 35 day-old guts, although female high-yeast guts had more mitoses, this difference was not significant in feminized males (*Figure 3—figure supplement 2B*). One possible explanation is that the observed pathology is an accumulation of ISC activity, which could expose subtle differences in mitotic rate over a lifetime. Another possibility is that ISC division and pathology are linked but pathology is driven by other factors that are responsive to diet, such as changes to the microbiota (*Clark et al., 2015*; *Petkau et al., 2014*). The TOR pathway inhibitor rapamycin extends lifespan in females, and to a much lesser extent in males (*Bjedov et al., 2010*), and a recent study showed that it decreases ISC division and slows barrier dysfunction in females (*Fan et al., 2015*). When we treated males, females and feminized males with rapamycin, we found that ISC proliferation and gut pathology were reduced in females and feminized males (*Figure 3N*). This shows that, similarly to DR, females derive a greater benefit from rapamycin than do males, possibly explaining, at least in part, the sex bias in the magnitude of lifespan extension by the drug.

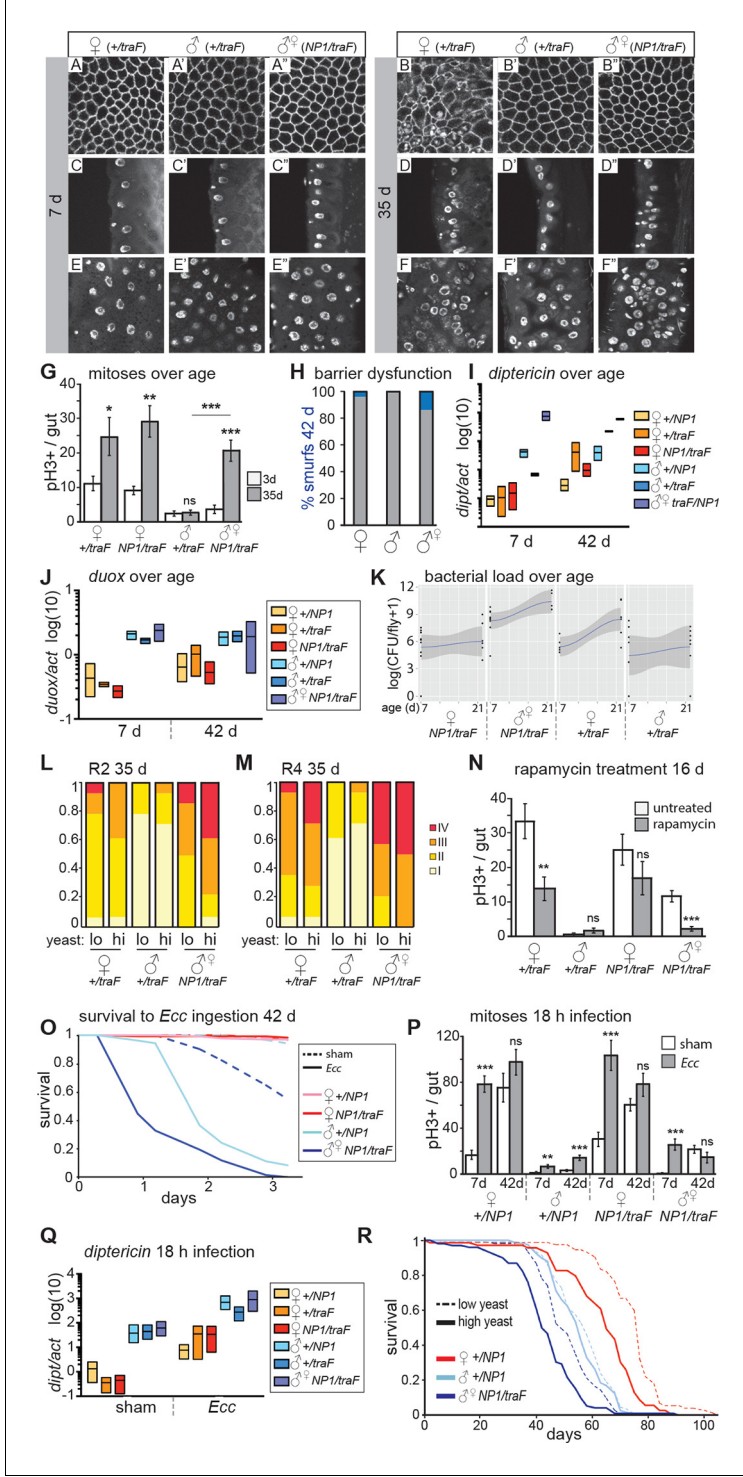

**Figure 3.** Feminized male guts develop female-like intestinal pathologies. (A-G) Mis-expression of *tra^F* feminizes male midguts. (A-F") PV (A-B") and R2 (C-F") morphology in *+/tra^F* females (A-F), *+/tra^F* males (A'-F') and *NP1>tra^F* males (A"-F") at 7 and 35 days, reveal female-like pathology in the R2 region of *NP1>tra^F* males at 35 days (F"). The *NP1* driver is not expressed in the majority of the PV and accordingly, the PV is well-maintained at 35 days (B"). Control females and feminized males increased ISC proliferation over age, but control males did not (n=10–20 guts per condition, student's *t* test, p=0.0366 for 3 vs 35 day-old *+/tra^F* females, p=0.0015 for 3 vs 35 day-old *NP1>tra^F* females, p=0.7057 for 3 d vs 35 d *+/tra^F* males, p=0.00022 for 3 vs 35 day-old *NP1>tra^F* feminized males). Feminized male guts (*NP1>tra^F*) had more mitoses at 35 days than control (*+/tra^F*) male guts (p=0.00018)
*Figure 3 continued on next page*

*Figure 3 continued*

(**G**). (**H-J**) barrier dysfunction and systemic AMP expression were increased in feminized males. Barrier dysfunction was significantly higher in feminized males than control (+/tra$^F$) males at 42 days (n≥150 per group, Fisher's exact, p=0.0001) and control (+/tra$^F$) females (p=0.0002) (**H**). *Diptericin* expression was increased over aging in all genotypes (n≥3 samples per condition, 10 individuals pooled per sample, 2 technical repeats; 2-way ANOVA, age p=0.0487, condition p=0.1031, interaction p=0.3485) and was increased in feminized males relative to control males at 7 days only (*t* test with Welch's correction, p=0.0018 for *NP1>tra$^F$* vs *+/NP1* at 7 days; p=0.5152 for *NP1>tra$^F$* vs *+/NP1* at 42 days; p=0.0011 for *NP1>tra$^F$* vs *+/tra$^F$* at 7 days; p=0.8907 for *NP1>tra$^F$* vs *+/tra$^F$* at 42 days) (**I**). *Doux* expression did not increase over aging in any genotype, but was higher in males than females overall (**J**). (**K**) Aerobic bacterial load tended to increase between 7 and 21 days for both sexes and genotypes (n≥8 samples per condition, 5 individuals pooled per sample; Monte Carlo Markov Chain Generalised Linear Model with Poisson Error Family, where *pMCMC*=0.040 for males and *pMCMC*=0.064 for females). Feminized males had a significantly higher load than control males (*pMCMC*<0.001). In addition, the direction of bias compared to females was switched in feminized males, such that control males had lower load than females, but feminized males had a higher load. A similar result was obtained for anaerobic load. (**L-N**) Pathologies in feminized males are responsive to diet and rapamycin treatment. Pathologies were binned into scaled categories and quantified, n≥12 per condition. PV categories as described in *Figure 2* legend (see *Figure 3—figure supplement 2* for PV scoring). R2 and R4 categories were defined as follows: I = WT, single layer epithelium with low number of basal ISCs. II = sporadic pathology of small nuclei 'nests' without significant disruption to the epithelium; III = widespread pathology, majority of epithelium has several layers of nuclei; IV = widespread pathology plus clear tumor formation. Gut feminized males have significantly worse pathology than control males on both diets in R2 (OLR, low-yeast, z=-3.916, p=0.0000899; high-yeast z=-4.339, p=0.0000143) and R4 (low-yeast, z=-4.012, p=0.0000602; high-yeast z=-4.520, p=0.0000617). The incidence of severe pathology and tumors (cat IV) in R2 was greater in feminized males than control females on high yeast diet (p=0.04) but not low yeast diet (p=0.48), suggesting that there was a cost of feminization that was partly alleviated by DR (**L-M**). Rapamycin treatment decreased mitoses in females and feminized males at 16 days (n≥10 guts per condition, students *t* test; control (+/tra$^F$) females, p=0.0079; control (+/tra$^F$) males, p=0.1; control females (*NP1/tra$^F$*), p=0.22; feminized males (*NP1/tra$^F$*), p=0.0001). (**N**) **O-R** Feminized males were more sensitive to oral infection, but acquired a lifespan response to dietary restriction. At 42 days males succumbed to *Ecc* oral infection while females did not. Feminized males died significantly sooner than controls (**O**). After *Ecc* oral infection at 7 days, males and females of all genotypes increased gut mitoses compared to sham infected (n≥10 guts per condition, students *t* test; control (+/NP1) females, p=2.082E-06; control (+/NP1) males, p=0.0011; control females (*NP1/tra$^F$*), p=0.00017; feminized males (*NP1/tra$^F$*), p=0.00045). However, females and feminized males lost the response to infection against a background of high proliferation in unchallenged individuals at 42 days (n≥10 guts per condition, students *t* test; control (+/NP1) females, p=0.2; control (+/NP1) males, p=0.0088; control females (*NP1/tra$^F$*), p=0.1478; feminized males (*NP1/tra$^F$*), p=0.2344) (**P**). Systemic *dipt* expression was increased after 18 hr continuous infection in all genotypes at 42 days (n≥10 guts per condition, *t* test with Welch's correction; *+/NP1* females, p=0.0571; *+/NP1* males, p=0.0132; *+/tra$^F$* females, p=0.0376; *+/tra$^F$* males, p=0.0282; *NP1/tra$^F$* females, p=0.0110; *NP1/tra$^F$* feminized males p=0.0331), but at a higher level in males than females in both sham and infected conditions (sham: *+/NP1* females vs males, p=0.0135; *+/tra$^F$* females vs males, p=0.0428; *NP1/tra$^F$* females vs males, p=0.0022. Infected: *NP1/+* females vs males, p=0.0012; *+/tra$^F$* females vs males, p=0.0964; *NP1/tra$^F$* females vs males, p=0.0237.) (**Q**). Lifespan analysis of *NP1>tra$^F$* males and *+/NP1* control males and females on two yeast dilutions. *NP1>tra$^F$* males were significantly shorter lived than control males on both standard (low yeast; log rank, p=0.0023) and double (high yeast; log rank, p=2.06E-11) yeast dilutions, whereas *+/NP1* control males did not differ between food conditions (log rank, p=0.34). This is a representative lifespan of three with similar outcomes. Cox proportional hazards analysis of the lifespan demonstrated a significantly increased risk of dying on high-yeast vs low-yeast food overall (p=2 x 10–16), and a significant difference in the response to food between control male genotypes and *NP1>tra$^F$* (gut feminized) males (p=0.0298). For full analysis, see *Figure 3—source data 1*. PV, proventriculus.

The following source data and figure supplements are available for figure 3:

**Source data 1.** Output table for Cox Proportional Hazards analysis of the *NP1>tra$^F$* (feminized gut) lifespan (*Figure 3Q*), showing hazard ratios, z and p values, and significance for all interactions.

**Figure supplement 1.** Feminization by misexpression of *tra$^F$*.

**Figure supplement 2.** Feminized males increase mitoses but do not resist oral *Ecc* infection.

## Feminized males are not more robust to oral infection, but their lifespan responds to DR

To assess whether the acquired ISC activity in gut-feminized males could contribute to a more robust survival to intestinal stress, we challenged them to *Ecc* oral infection at 7 and 42 days of age. At 5-days post-infection, 7-day-old females and males survived, but approximately 50% of gut-feminized males had died (*Figure 3—figure supplement 2D*). Analysis of ISCs post-infection showed that feminized males induced significantly higher proliferation than control males, but this was clearly not protective (*Figure 3P*). In aged (42 day-old) flies, as before, we found that only males succumbed to *Ecc* oral infection. Feminized males were most susceptible, dying more quickly than did control males (*Figure 3O*, *Figure 3—figure supplement 2E*), and this was not related to a change in feeding on either normal food (*Figure 3—figure supplement 2F*) or the infection pellet (*Figure 3—figure supplement 2G*). At this age, female guts had a high number of mitoses, but these did not significantly increase on infection, and nor did they in feminized males (*Figure 3P*). One interpretation of this result is that, in aged females and feminized males, the pool of ISCs has been exhausted by continued mitotic activity through the lifespan, or that due to age-related triggers such as inflammation or dysbiosis, ISC activity is already at a maximum. Markedly different to females, however, was the high induction of systemic *dipt* in control and feminized males (*Figure 3Q*), in line with their reduced survival. Altogether, these data show that age-related sensitivity of males to intestinal infection was not rescued by feminization of the midgut. One probable contributing factor to the high death rate of feminized males is the barrier dysfunction observed at 42 days (*Figure 3H*). However, their sensitivity to *Ecc* at young ages points to other, compromised immune responses. We hypothesize that a sex mis-match between the gut and other immune tissues such as the fat body and hemocytes, may be detrimental.

Females show a greater lifespan extension than do males when subjected to DR (*Magwere et al., 2004*). Males from the outbred line $w^{Dah}$ used in this study have a mean lifespan that is 22 days shorter than that of their female siblings' (*Figure 2—figure supplement 2D*), and only females showed a lifespan extension when raised on food with a 50% reduction in yeast (*Figure 2—figure supplement 2D*). Comparison of the lifespans of gut-feminized males with control female and male flies raised on a standard diet showed that gut-feminized males were significantly shorter lived than control males (*Figure 3R*, *Figure 3—figure supplement 2H*), suggesting that their acquired intestinal hyperplasia was detrimental to their lifespan. Interestingly, when we measured the responses of lifespan to two different yeast dilutions, we found that midgut-feminized males had acquired a response to DR similar in magnitude to that seen in control females (*Figure 3R*, *Figure 3—figure supplement 2H* and *Figure 3—source data 1*). Taken together with our data on tissue sex and gut pathology, this finding suggests that lowered levels of intestinal pathology contribute to increased female longevity upon DR, and offers one explanation for the difference between males and females in the response of their lifespans to diet.

## Discussion

In marked contrast to the gut pathology previously reported in aging females (*Biteau et al., 2008*; *Choi et al., 2008*; *Patel et al., 2015*), and the extensive deterioration seen in multiple regions of the gut in the present study, we found that males generally have well-maintained guts into old age, even at ages close to the maximum male lifespan. Female intestinal pathology is driven by stem cell activity (*Patel et al., 2015*), which is responsive to diet (*Choi et al., 2011*; *O'Brien et al., 2011*). Gut hyperplasia is limiting for female lifespan: when ISC division is genetically reduced, female lifespan is extended (*Biteau et al., 2010*; *Hur et al., 2013*; *Rera et al., 2011*; *Ulgherait et al., 2014*). We showed that, by switching the sexual identity of male midgut cells, ISC activity was increased, pathologies appeared and lifespan was shortened. In addition, the lifespan of males with feminized guts became responsive to diet, with DR both reducing the risk for developing severe pathology and increasing lifespan. The marked sexual dimorphism in gut pathology during aging provides a potential explanation for why males do not increase lifespan as much as do females in response to DR. Because males do not suffer from age-related hyperplasia, they do not derive the same benefit from DR-induced stem cell quiescence. Limited hyperplasia could also at least partly explain the much greater increase in lifespan seen in females than in males upon reduced IIS (*Clancy et al., 2001*;

*Tatar et al., 2001*); given that ISC activity in females is responsive to insulin signaling (*Biteau et al., 2010*; *O'Brien et al., 2011*; *Hur et al., 2013*).

Intestinal barrier function decreases during aging in females, and this decline is associated with an increase in immune activation, a decrease in nutrient absorption and death (*Rera et al., 2012*). We showed that sustained ISC division has drastic consequences for the female gut epithelium, causing scars and holes to appear on the basal side. Interestingly, in transcriptomic analysis of aging guts of flies raised in both natural and axenic conditions, genes involved in wound healing were increased during aging (*Guo et al., 2014*). In line with the absence of severe pathology in males, barrier function was maintained into old age. Despite this, males succumbed to oral infection and xenobiotic stress to which females were resistant, and this sensitivity was not rescued by feminization. We found that the gut ISC response to infection, a repair process, was significantly lower in male flies and increased in feminized males. Regulation of ISC activity is therefore not the only difference between males and females affecting resistance: other epithelial immune responses such as ROS and antimicrobial peptide production or tolerance to *Ecc*-derived toxins could also be sexually dimorphic (*Vincent and Sharp, 2014*). The male sensitivity and sepsis-like response to *Ecc* oral infection, age-related systemic inflammation and high levels of ROS production shown in this study are striking and present a starting point for the analysis of sex differences in intestinal immunity, which will be relevant for resistance to opportunistic infection and shaping the microbiome.

Intestinal plasticity is important not only to resist infection (*Buchon et al., 2009*), but also to respond predictively to the increased energy demands of egg production upon mating (*Reiff et al., 2015*), which could explain why females and males invest so differently in ISC activity. Male reproduction does not require the same amount of nutritional input and, therefore, large-scale intestinal remodeling would probably not increase reproductive rate, which is promoted instead by increased detection and courtship of females (*Bretman et al., 2011*). This sex difference is evolutionarily conserved, in that female mammals extensively grow and remodel their guts, increasing digestive and absorptive capacity in accordance with the increased nutritional demands of lactation (*Hammond et al., 1997*; *Speakman et al., 2008*). Possibly related to this greater gut plasticity, women have higher rates of region-specific colon cancer than do men (*Kim et al., 2015*; *Jemal et al., 2011*), although the mechanisms underlying this difference are not well understood. Female-specific intestinal remodeling during reproduction could be a driving factor. Intriguingly, higher parity and earlier childbearing are protective against colon cancer, whereas later child-bearing is a significant risk factor (*Wernli et al., 2009*), suggesting that the timing of reproductive events may affect ISC (mis-)regulation. Further studies linking early gut plasticity, nutrition and fecundity with severity of hyperplastic pathology during aging in *Drosophila* may help elucidate conserved mechanisms underlying sex differences in cancer rates.

## Materials and methods

### Fly strains

All mutants were backcrossed for six generations into the wild type, outbred strain, $w^{Dahomey}$ ($w^{Dah}$), maintained in population cages. The following fly stocks were obtained from the Bloomington Stock Centre: *P{UAS-tra.F}20J7*, *P{GawB}Myo31DF^{NP0001}* (referred to as *NP1-Gal4*), and *daughterless-Gal4. P{PTT-un1}CG8668^{117-2} (Resille-GFP)* was originally obtained from the Flytrap project and was a gift from A Jacinto.

### Fly husbandry

Stocks were maintained and experiments conducted at 25°C on a 12 hr:12 hr light:dark cycle at 60% humidity, on food containing 10% (w/v) brewer's yeast, 5%(w/v) sucrose, and 1.5% (w/v) agar unless otherwise noted (referred to as 'low yeast' food denoting an arbitrary 1:1 sugar to yeast ratio, which is changed to 1:2 (20% yeast, 5% sucrose) for 'high yeast' food). For measurement of gut histology and lifespans, flies were reared at standard larval density and eclosing adults were collected over a 12-hr period. Flies were mated for 48 hr before being sorted into experimental vials at a density of 10 flies per vial. Flies were transferred to fresh vials thrice weekly, and for lifespan, deaths/censors were scored during transferral.

## Imaging of gut pathology

Guts were dissected from live flies in ice-cold PBS and immediately fixed in 4% formaldehyde for 15 min. Guts were mounted in mounting medium containing DAPI (Vectastain) and endogenous GFP was imaged immediately. Between 6 and 14 guts were analyzed per condition, per time point. Images were captured with a Zeiss (UK) LSM 700 confocal laser scanning microscope using a 40x oil-immersion objective.

## Immunohistochemistry

The following antibodies were used for cell division analyses; primary antibodies: rabbit anti-PH3 (Cell Signaling (Danvers, MA) 9701) 1:500; mouse anti-GFP (Cell Signaling 2955) 1:1000. Secondary antibodies: Alexa Fluor 594 donkey anti-rabbit ((A21207) Thermo Fisher Scientific, Waltham, MA) 1:1000; Alexa Fluor 488 donkey anti-mouse (A21202) 1:1000. Guts were dissected in ice cold PBS and immediately fixed in 4% formaldehyde for 15 min, serially dehydrated in MeOH, stored at -20°C, and subsequently stained. Guts were washed in 0.2% Triton-X / PBS, blocked in 5% bovine serum albumin / PBS, incubated in primary antibody overnight at 4°C and in secondary for 2 hr at RT. At least 10 guts per condition were mounted, scored and imaged as described above.

## Gut barrier analysis (Smurf assay)

Gut barrier efficiency was analyzed by placing flies on blue food (minimum 110 flies per condition, except at 80 d, min. 80 flies) prepared using using 2.5% (w/v) FD&C blue dye no. 1 (Fastcolors) as previously described (*Rera et al., 2012*), except flies were kept on the blue food for 24 hr before the Smurf phenotype was scored.

## Oral infection survival assay

At least 3 x colonies of *Ecc15* were grown in separate overnight cultures in Luria-Bertani medium, pooled, pelleted and adjusted to $OD^{600}$ = 200 with fresh LB. Bacteria were mixed 1:1 with 5% sucrose (final $OD^{600}$ = 100); LB / sucrose was used for sham infection. Flies were starved for 5 hr in empty vials. Experimental vials were lined with 10 pieces of Whatman filter paper to which 1 ml *Ecc15* / sucrose (or LB / sucrose) was added. Starved flies were added at a density of 20/vial; a minimum of 80 flies/condition. Filter paper was refreshed each day with newly grown *Ecc15* / sucrose (or LB / sucrose) solutions, using a Pasteur pipette. Deaths were scored thrice daily.

## DDT ingestion survival assay

Ten flies per vial, at least 110 flies per condition, were fed with the organochloride dichlorodiphenyl-trichloroethane (DDT; Supelco, Sigma-Aldrich, UK) for 18 hr. To standard SYA food, 0.03% (w/v) DDT, made from a stock solution of 2% (w/v) DDT in ethanol, was added. Deaths were scored thrice daily.

## Feeding assay

Flies were transferred to vials at a density of 5 per vial on the evening before the assay. Vials were coded and placed in a randomized order on viewing racks at 25°C overnight. The assay occurred with minimal noise and physical disturbance. Thirty minutes was allowed between the arrival of the observer and commencement of the assay. Observations were performed blind for 90 min, 1 hr after lights-on. Vials were observed for approximately 3 s during which the number of flies feeding was noted. A feeding event was scored when a fly extended its proboscis and made contact with food. Successive rounds of observations were carried out giving an observation rate of once / 5 min. Feeding data are expressed as a proportion by experimental group (sum of scored feeding events divided by total number of feeding opportunities, where total number of feeding opportunities = number of flies in vial×number of vials in the group×number of observations). For statistical analyses, comparisons between experimental groups were made on total feeding events by all flies within a vial, to avoid pseudoreplication.

## Bacterial load measurements

As described for guts (*Broderick et al., 2014*), whole flies were surface sterilized for 1 min in 95% ethanol, rinsed and pooled into groups of 5, $n \geq 8$ per condition, on ice for homogenization.

Three 1/5 serial dilutions were plated on Man, Rogosa and Sharpe (MRS) plates and cultured at 29°c both aerobically and anaerobically, and scored at 48 and 72 hr.

## Gut cell size measurements

Nearest-neighbour internuclear distance in the R2 region was measured from raw LSM files (Zeiss) using the Measure function in Fiji (Image J); 20 distances per gut, $n \geq 6$ guts per condition.

## RNA isolation and quantitative RT-PCR analysis

RNA was isolated from $n \geq 14$ guts per sample with TRIzol (Invitrogen, Thermo Fisher Scientific, Waltham, MA) and cDNA was synthesized by using SuperScript II Reverse Transcriptase (Invitrogen) according to manufacturers instructions. qPCR was carried out by using Power SYBR Green PCR Master Mix (Thermo Fisher Scientific) on QuantStudio6 Flex real-time PCR (Applied Biosystems, Thermo Fisher Scientific, Waltham, MA). The following primer sequences (Eurofins, UK) were used in the analysis: *dsxF*-F (TCAACACGTTCGCATCACAAA); *dsxF*-R (TAGACTGTGA TTAGCCCAATAACTGA), *act5C*-F (CACACCAAATCTTACAAAATGTGTGA); *act5C*-R (AATCCGGCC TTGCACATG); *dipericin*-F (GCGGCGATGGTTTTGG); *dipericin*-R (CGCTGGTCCACACCTTCTG); *Duox*-F (TAGCAAGCCGGTGTCGCAATCAAT); *Duox*-R: ACGGCCAGAGCACTTGCACATAG.

## Statistical analyses

Statistical analyses were performed in Excel (Microsoft) or Prism (Graphpad, La Jolla, CA), except for Ordinal Logistical Regression (OLR), Cox Proportional Hazards and Monte Carlo Markov Chain Generalised Linear Model with Poisson Error Family analyses, which were performed in R (using the clm function from the 'ordinal' library for OLR). Statistical tests used are indicated in the figure captions. Outliers were rarely excluded from the data, but when excluded conformed to the rule of lying more than two standard deviations away from the mean. They are clearly marked as data points on graphical displays.

## Acknowledgements

We thank the reviewing editor and reviewers for insightful comments and suggestions that greatly improved the manuscript. We thank N Alic for discussion and help with statistical analyses, N Woodling for graphics and help with statistical analyses, and A Zaidman-Remy for comments on the manuscript. We also thank M Ahmad and G Vinti for technical assistance, and the Partridge lab for discussion and support. The *Resille-GFP* epithelial reporter line was kindly provided by A Jacinto. Stocks from the Bloomington Drosophila Stock Center (NIH P40OD018537) were used in this study.

## Additional information

### Funding

| Funder | Grant reference number | Author |
|---|---|---|
| FP7/2007-2013/ ERC | 259679 - IDEAL | Jennifer C Regan<br>Linda Partridge |
| The Max Planck Institute | | Mobina Khericha<br>Linda Partridge |
| Wellcome Trust | Strategic Award WT098565AIA | Mobina Khericha<br>Ekin Bolukbasi<br>Linda Partridge |
| FP7/2007-2013 / ERC | 268739 | Linda Partridge |
| The Royal Thai Government | | Nattaphong Rattanavirotkul |

The funders had no role in study design, data collection and interpretation, or the decision to submit the work for publication.

## Author contributions

JCR, Conception and design, Acquisition of data, Analysis and interpretation of data, Drafting or revising the article; MK, Conception and design, Acquisition of data, Analysis and interpretation of data; AJD, EB, NR, Acquisition of data, Analysis and interpretation of data; LP, Conception and design, Analysis and interpretation of data, Drafting or revising the article

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
