## [Decision Letter]

Thank you for submitting your work entitled "Sex difference in pathology of the ageing gut mediates the enhanced response of female lifespan to dietary restriction" for peer review at *eLife*. Your submission has been favorably evaluated by Sean Morrison (Senior editor) and three reviewers, one of whom, Andrew Dillin, is a member of our Board of Reviewing Editors, and another is David Walker.

The reviewers have discussed the reviews with one another and the Reviewing editor has drafted this decision to help you prepare a revised submission.

The physiological differences to explain lifespan differences among the sexes is poorly understood. Using the fruit fly *Drosophila* melanogaster, Regan and colleagues correlate certain aspects of gut plasticity to fly aging in males vs. females. Increased intestinal stem cell division found in females is correlated with shorter lifespan compared to the division rate in males. By converting male intestines to a female state, the authors can shorten male lifespan and change the response to dietary restriction.

In general, the reviewers appreciated the work and potential impact of the work. However, there was unanimous agreement that better analysis of the gut during aging is needed as well as analysis of other parameters of gut functions, such as anti-microbial defense systems and antioxidant systems.

The following major points were found across all three reviews:

1) More careful analysis of gut function breakdown during aging is required. While all reviewers agree with the analysis of ISC proliferation, more detailed and later timepoints need to be performed, especially in the case of male lifespan. Please consider Reviewer #3’s statements about other correlates.

2) Do the microbiota change in males vs. females, especially given the genetic alteration of male flies with a feminized gut. While it is agreed that a complete study of the microbiota is beyond the scope of this study, could a cursory analysis be completed to test if gross changes are observed? The same holds for dysbiosis.

3) Data is provided for the lifespan effects on feminized guts of males, but infection by *Ecc* is not. Please test. Additionally, because this is a lifespan shortening effect, the authors need to control for strain sickness. One idea is to test another longevity paradigm that is not sex modified.

4) Throughout the text references are lacking or misplaced and prior work in this area is not discussed sufficiently.

5) Appropriate driver controls for all experiments need to be presented.

Reviewer #1:

In general this is a very interesting set of observations that corroborate several observations in the fly aging field. Sex specific differences in fly lifespan have been reported, yet a causal mechanism has been elusive. Here, the authors have investigated gut "function" during aging and find that female guts tend to attempt repair and proliferation more than male guts. While females are more resistant to toxins ingested (such as *Erwinia carotovora* infection), their guts deteriorate faster. Accordingly, diet restriction appears to preserve gut function in aged female flies. In male flies, feminizing parts of the intestine creates dysplasia and now makes the males responsive to DR, albeit the transgenic males are already short-lived.

Overall, the study is well performed and the interpretation of the existing data is good. However, there are a few points that should be addressed.

The issue of microbiota playing a role in males vs. females should be analyzed. Is there a huge difference and can this be reversed in the feminized males?

The lifespan analysis: showing that DR can make a sick strain more healthy is nice, but is it specific as stated? Is there a different lifespan extending paradigm that is not sex specific that can be tested? If so, does it increase longevity of the feminized animals?

Some analysis of the feminized animals has been carefully performed. One piece lacking is the resistance to *Ecc* infection or other toxin (DDT) treatment. Are the feminized males now resistant?

Reviewer #2:

In general, I find the paper and findings interesting. The physiological origins of sex differences in longevity are poorly understood. The major strength of the paper is the data showing that male flies with feminized guts show 'female-like' pathology and show an improved lifespan response upon DR. My major concern, however, is that some of the claims of the paper are overstated and not fully supported by the data. Another issue regards the novelty of the findings and, perhaps more importantly, the authors not giving credit to previous work where credit is due.

Major issues:

Novelty:

The Edgar lab reported in Cell. 2009 Jun 26;137(7):1343-55 that male guts show greatly reduced turnover rates compared to females.

The Walker lab reported in PNAS 2012 109(55):21528-33 that diet restriction delays intestinal pathology in female flies.

In the current manuscript, the authors build upon and extend both of these findings. But, they do not cite the preceding work. Citing the previous work doesn't harm the current paper. In fact, it strengthens the claims of the paper.

Claims of the paper not supported by the data:

In both the Abstract and first line of the Discussion it is claimed that:

Abstract: "male flies die with their guts largely unaffected by ageing"

Discussion: "male flies maintain a phenotypically perfect gut until death".

However, this claim is not supported by the data in paper. In order to support such a claim, the authors would need to examine gut pathology in a large number of male flies in the days (hours?) preceding death. Referring to Figure 2, in the text, the authors state that 'many' male flies show no intestinal pathology up to 70d of age. I am missing this data. In the figures, I only see data up to 42 days of age.

How many male flies were tested? What ages? How many 70d old male flies showed 'gut pathology'? How many didn't? Statistics?

It appears that intestinal barrier function was assayed at 42d and no male 'smurfs' were observed. However, in order to test the idea that male flies (of this strain) do not show barrier failure prior to death, a large number of older (70d as in other assays?) male flies should be examined. This is important because male flies of another laboratory strain (*w^1118^*) have been reported to show age-onset intestinal barrier dysfunction. See Rera et al. PNAS 2012 (Supp material). I appreciate that the authors are working with an outbred WT line and perhaps differences in age-onset pathology exist between lab strains. But, again, this earlier work (Rera PNAS 2012) should be cited in this context and discussed if indeed strain differences exist.

To be clear, the authors' data supports the idea that male guts show a *delay* in the onset of 'gut pathology'. But, if the authors wish to make the further/stronger claim that male flies, of their lab strain, do not show *any* (or negligible) age-onset gut pathology they need to provide more data.

The *Ecc* infection data is potentially interesting. But, I don't find the data compelling evidence that male flies show reduced tolerance to infection due to reduced 'gut plasticity' as suggested in the abstract. There are a number of potential confounding factors that are not addressed:

1) The flies were starved prior to the assay. This may lead to male/female differences in feeding. It is likely that male flies are more sensitive to starvation than females. Perhaps after a short period of starvation male flies eat more than females. If so, this would be a confounding factor, i.e. male flies are more sensitive to the *Ecc* infection because they take in more during the oral infection. Can the authors exclude this possibility?

2) Also, it appears that the assay was carried out at 35d of age. As female flies live longer than males it is difficult to interpret this finding. 35d old females will likely be 'healthier' than 35d old males. At 35 days of age, female flies will likely be more resistant to most extrinsic stressors, including stressors that do not act exclusively on the gut (e.g., heat, hyperoxia). Perhaps 35d old female flies have better immune function than 35d old males?

3) In Figure 2, it appears that male guts do show an ISC proliferation response to the infection. In fact, it could be argued that the magnitude of the male response is GREATER than that of the female response.

In summary, there are a number of reasons why at 35 days of age male flies are more sensitive to this infection. I would suggest that the authors are more careful not to overstate their findings and/or provide stronger evidence to support their claims.

Suggestions to improve impact

A potential role for the microbiota in sex differences in gut pathology is ignored. Several studies have reported that the presence of gut-associated microbes contributes to 'gut pathology' in female flies. More specifically, axenic female flies show reduced 'gut pathology' with age (Broderick et al., 2014, Buchon et al., 2009, Guo et al., 2014; Clark et al. 2015). This is, of course, what the authors claim to observe in the aging intestines of male flies. It would, therefore, be interesting to find out whether male flies show an age-related increase in bacterial load and/or microbial imbalance (dysbiosis) in the strains used in this study. If they do not, this may contribute to the delayed intestinal degeneration observed. I am not suggesting that this needs to be figured out in detail in the current manuscript. But, it is certainly worth investigating and could, potentially, provide insight. It would also be interesting and strengthen the claims if the authors could show that feminized male guts show alterations in the microbiota.

Reviewer #3:

This manuscript by Regan et al. explores sex-specific differences in longevity and gut pathology with age in *Drosophila*. The relationship between lifespan and the gut's integrity and ability to respond challenges modulated by sex and age is largely unexplored in mechanistic detail, but is of significant general interest. The authors describe the phenomenon of sexual dimorphic gut phenotypes with age in some detail, but underlying mechanisms of the relationship remains to be elucidated. However, much of the conclusions are based on (not yet given) answers to points 1-3 below.

1) Since the male gut maintains its integrity with age but live shorter the relationship to longevity is not clear, except that the superimposition of a 'feminization' of the gut in a male body seems to make things worse.

2) Given point 1, other measures of decline should be explored, such as oxidative stress, anti-microbial responsiveness (AMPs, etc.), proteostasis, etc., the authors allude to but do not consider, in order to provide additional direct (or inverse) correlates, other than ISC numbers. Regarding the latter, could the ISC number (or anti-microbial responsiveness) be directly reduced in the female gut or elevated in the male gut, other than (systemic) DR, that may have indirect effects?

3) The high male vs female sensitivity of males to bacterial infection or DDT is interesting but insufficiently explored to be conclusive. Do ISC proliferate more, produce a higher anti-microbial response in females? What happens in young females, are they more susceptible? The discrepancy between susceptibility and gut integrity/permeability is puzzling and in a way counterintuitive, despite the speculation that higher ISC numbers may provide better potential repair (not demonstrated!). What is the progression of gut and other pathologies upon infection or DDT? Does the gut disintegrate more quickly in males with these challenges, or is the resulting morbidity unrelated to the gut?

4) Controls are missing: Both driver and UAS controls should be provided for all experiments, not just one for longevity studies and the other for other experiments.

5) The high-low yeast should be better explained in terms of concentration and how this relates to 1/2SYA in lifespan studies.

---

## [Author Response]

*1) More careful analysis of gut function breakdown during aging is required. While all reviewers agree with the analysis of ISC proliferation, more detailed and later timepoints need to be performed, especially in the case of male lifespan. Please consider Reviewer #3’s statements about other correlates.*

We have responded to this in some detail, including analysis of barrier function and pathology at older ages, AMP expression and ROS production over ageing. Also, the analysis of the age-dependent response to gut infection has been expanded as described below.

*2) Do the microbiota change in males vs. females, especially given the genetic alteration of male flies with a feminized gut. While it is agreed that a complete study of the microbiota is beyond the scope of this study, could a cursory analysis be completed to test if gross changes are observed? The same holds for dysbiosis.*

We have analyzed bacterial load over age, in males, females and feminized males, and the proportions of internal bacteria that can be cultured aerobically and anaerobically, providing some interesting new data.

*3) Data is provided for the lifespan effects on feminized guts of males, but infection by Ecc is not. Please test. Additionally, because this is a lifespan shortening effect, the authors need to control for strain sickness. One idea is to test another longevity paradigm that is not sex modified.*

We have analyzed the response to *Ecc* infection in males, females and feminized males, in some detail, including survival, induction of AMPs and ROS, and stem cell activity. To respond to the concern of strain sickness, we have investigated the effect of rapamycin treatment on gut pathology in both sexes and feminized males, and have further analyzed our existing lifespans with additional statistical tests.

*4) Throughout the text references are lacking or misplaced and prior work in this area is not discussed sufficiently.*

This has been dealt with and we feel that the references now better reflect the field.

*5) Appropriate driver controls for all experiments need to be presented.*

Appropriate controls for experiments have been added to the manuscript and included in new analyses.

*Reviewer #1:[…] The issue of microbiota playing a role in males vs. females should be analyzed. Is there a huge difference and can this be reversed in the feminized males?*

We agree that identifying sex differences in the microbiota is interesting and informative. We have analysed internal bacterial load during ageing, and also the proportions of aerobic and anaerobic species, in males, females and feminized males. We found a higher bacterial load in females than in males at all ages examined. This correlates with the lower incidence (or lack of) of barrier dysfunction observed in *w^Dah^*males in this study. Strikingly, gut-feminized males had an increased load compared to both males and females. Internal bacterial load can thus be changed by feminization of gut tissue. In line with recent studies on females (Rera 2012, Guo 2014, Broderick 2014, Clark 2015) this increased bacterial load correlated with a high level of barrier dysfunction and systemic AMP expression in feminized males. This is now discussed in the manuscript. We also analyzed the proportions of aerobic and anaerobic species, an indication of the proportions of the two major genera (*Acetobacter* and *Lactobacilli*,), in 18 d and 40 d flies. We did not find significant differences in proportions either with ageing, or when comparing males, females and feminized males. A recent study by Clark et al. from the Walker lab demonstrated that changes in lower-abundance species, elucidated by deep sequencing of the microbiome, are important for age-related changes to the intestinal tissue in females. Our more general analysis will have missed subtle changes that could nevertheless have a significant impact on gut homeostasis. We have chosen not to present these data in the manuscript because we think that a much more detailed analysis would be required to draw any firm conclusions on sex differences in the composition of the microbiota and their potential impact.

*The lifespan analysis: showing that DR can make a sick strain more healthy is nice, but is it specific as stated? Is there a different lifespan extending paradigm that is not sex specific that can be tested? If so, does it increase longevity of the feminized animals?*

We are not sure what the question is here. If it is: ‘DR makes healthy female flies live longer too, so can we be sure that it is the rescue of gut pathology that induces the response of lifespan to DR in the gut-feminized males? Given that the only sex reversal in these males is to a specific gut region, it must be ultimately responsible for the other phenotypes seen in these flies. To test whether the feminization results in a full, female-like, response of lifespan to DR we have performed Cox Proportional Hazard statistical analyses on three repeat measurements of the lifespans of gut-feminized males. This anlaysis showed that gut feminized males derived the same degree of lifespan extension as that seen in females. This finding supports the idea that the lifespans are being extended by the same mechanisms.

To probe the generality of the role of rescue of gut pathology in sex differences in response to lifespan-extending interventions, we analysed the guts of rapamycin-treated flies during ageing. Like DR, rapamycin extends lifespan in both males and females, but to a greater extent in females (Bjedov 2010). We find that epithelial pathology is delayed in rapamycin-treated females and, in addition, females have lower numbers of ISCs after treatment, in line with a recent study on rapamycin treatment and gut ageing (Fan 2015). Males do not derive an obvious change in gut maintenance or ISC proliferation with rapamycin treatment. Although we did not have time to analyse the lifespan of gut-feminized males on rapamycin, we have qualitatively analyzed gut pathology and quantified stem cell activity in these flies. Strikingly, ISC division is significantly reduced in females and feminized males, but not in males, after 2 weeks of rapamycin treatment. These new data show that, under another lifespan extending treatment (rapamycin) where males and females have a differential longevity response, the effect on gut maintenance is greater in females, who show a greater decline, and that this pathology is rescued in feminized males by the treatment. This further supports the idea that rescue of gut maintenance is a driver for extended lifespan in females and gut-feminized males.

*Some analysis of the feminized animals has been carefully performed. One piece lacking is the resistance to Ecc infection or other toxin (DDT) treatment. Are the feminized males now resistant?*

We agree that this is an important piece of information. We have subjected gut-feminized males to *Ecc* oral infection at young (7 d) and old (42 d) ages, and analysed survival and the ISC response. In young flies, only feminized males were sensitive to the infection. Females, males and feminized males all significantly increased mitoses after infection, but females had a far higher number of active ISCs per se. ISCs in feminized males proliferated more than in control males, but this was clearly not protective against the infection. In aged flies, in line with our data comparing males and females, only males died from *Ecc* infection. Feminized males were again the most susceptible to oral infection, dying sooner than both control male genotypes. Infected males had significantly more mitoses than sham infected individuals. Females, however, had a high number of mitoses, which did not increase significantly on infection. Similarly, feminized males did not increase ISC proliferation upon infection. One interpretation of this result is that in aged females and feminized males, the pool of ISCs has been exhausted by continued mitotic activity through the lifespan. Another is that, due to age-related triggers of stem cell activity such as inflammation or dysbiosis, stem cell activity is already at a maximum. These data show that age-related sensitivity of males to intestinal infection is not rescued by feminization of the midgut. One probable contributing factor to the high death rate of feminized males is their increased barrier dysfunction at old age (see below). However, their sensitivity to *Ecc* at 7 d points to other compromised immune responses. One possibility is that a sex mis-match between the gut and other tissues involved in the immune response to oral infection is detrimental.

*Reviewer #2: In general, I find the paper and findings interesting. The physiological origins of sex differences in longevity are poorly understood. The major strength of the paper is the data showing that male flies with feminized guts show 'female-like' pathology and show an improved lifespan response upon DR. My major concern, however, is that some of the claims of the paper are overstated and not fully supported by the data. Another issue regards the novelty of the findings and, perhaps more importantly, the authors not giving credit to previous work where credit is due. Major issues: Novelty: The Edgar lab reported in Cell. 2009 Jun 26;137(7):1343-55 that male guts show greatly reduced turnover rates compared to females. The Walker lab reported in PNAS 2012 109(55):21528-33 that diet restriction delays intestinal pathology in female flies.*

Both these studies are now cited. We consider that the novelty of the study is the thorough, new documentation of the sex difference in gut pathology during ageing, and the demonstration that by feminizing a region of the gut we can induce a response of gut pathology and lifespan to DR, thus explaining the sexually dimorphic response to this intervention. We also now present data suggesting that the same may be true of the sexually dimorphic response of lifespan to rapamycin.

*In the current manuscript, the authors build upon and extend both of these findings. But, they do not cite the preceding work. Citing the previous work doesn't harm the current paper. In fact, it strengthens the claims of the paper.*

We had missed both findings – it was not a deliberate omission – and in fact citing these papers strengthens the repeatability of aspects of our own study

*Claims of the paper not supported by the data: In both the Abstract and first line of the Discussion it is claimed that: Abstract: "male flies die with their guts largely unaffected by ageing"Discussion: "male flies maintain a phenotypically perfect gut until death".However, this claim is not supported by the data in paper. In order to support such a claim, the authors would need to examine gut pathology in a large number of male flies in the days (hours?) preceding death. Referring to Figure 2, in the text, the authors state that 'many' male flies show no intestinal pathology up to 70d of age. I am missing this data. In the figures, I only see data up to 42 days of age. How many male flies were tested? What ages? How many 70d old male flies showed 'gut pathology'? How many didn't? Statistics?*

We agree that analysis of very old flies is lacking from the manuscript and that claims of reduced/absent pathology should be supported by such data. To address this, we have included both qualitative and quantitative data on older male flies, at 64 d of age (which is approx. equal to the median lifespan of *w^Dah^* males). We agree that, because some males at old ages show the beginnings of gut pathology, it is more accurate to describe males as having a ‘delayed’ decline in gut structure. To clarify our findings, we have described in detail our categories for scoring pathology, and the difference between distinct gut regions. For example, at 64 d, the strongest sex bias we find is in the pathology of the PV, where females show a spectacular decline, whereas males show very little. In R2 and R4, around half of 64 d males show a ‘prepathology’; that is sporadic, small clusters of stem cells/enteroblasts (category II). These do not have an effect on the overall epithelial structure, but precede the appearance of widespread small tumours (category III) and large tumours (category IV) that disrupt the integrity of the epithelium. A small percentage (~10%) of males show a more significant pathology (category III). We never found large tumours (category IV) in males, in any gut region. We have also assessed the barrier function of males and females at 80 d, as described below.

*It appears that intestinal barrier function was assayed at 42d and no male 'smurfs' were observed. However, in order to test the idea that male flies (of this strain) do not show barrier failure prior to death, a large number of older (70d as in other assays?) male flies should be examined. This is important because male flies of another laboratory strain (w^1118^) have been reported to show age-onset intestinal barrier dysfunction. See Rera* et al. *PNAS 2012 (Supp material). I appreciate that the authors are working with an outbred WT line and perhaps differences in age-onset pathology exist between lab strains. But, again, this earlier work (Rera PNAS 2012) should be cited in this context and discussed if indeed strain differences exist.*

We have looked for an onset of barrier dysfunction in male flies with age, by performing Smurf analyses at 80 d, in *w^Dah^* males and females. We find that, even in this oldest cohort, we never observe male Smurf flies appearing in this line. In parallel, we tested the *w^1118^* line for loss of barrier function, and we found that, similarly to the Walker lab (although at a lower rate than reported in Rera 2012), a small proportion of 42 d old males become Smurfs, although significantly fewer than females. Given the recent data from Clark 2015 demonstrating the tight link between microbiota and dysfunction, we find it unsurprising that different labs would observe differing Smurf rates. In addition, when we feminize male guts, a significant proportion show barrier dysfunction at 42 d, at a higher rate than control females. We think these data strengthen the idea that sex-specific rates of gut function decline exist and that barrier dysfunction, while being critical for female lifespan, may not be limiting to male lifespan.

*To be clear, the authors' data supports the idea that male guts show a delay in the onset of 'gut pathology'. But, if the authors wish to make the further/stronger claim that male flies, of their lab strain, do not show any (or negligible) age-onset gut pathology they need to provide more data.*

In addition to the new data described above, we have taken your advice and described male gut ageing more carefully.

*The Ecc infection data is potentially interesting. But, I don't find the data compelling evidence that male flies show reduced tolerance to infection due to reduced 'gut plasticity' as suggested in the abstract. There are a number of potential confounding factors that are not addressed: 1) The flies were starved prior to the assay. This may lead to male/female differences in feeding. It is likely that male flies are more sensitive to starvation than females. Perhaps after a short period of starvation male flies eat more than females. If so, this would be a confounding factor, i.e. male flies are more sensitive to the Ecc infection because they take in more during the oral infection. Can the authors exclude this possibility?*

Good point. We have measured feeding rate of male, female and gut feminized male flies, on normal food (2 different yeast concentrations were included), and on starved, infected flies (fed bacterial pellet/sucrose or LB/sucrose) following the same feeding protocol as for infection survival assays. We found that on normal SYA food, males eat significantly less than females. Interestingly, gut-feminized feeding rates are the same as control males, suggesting that their feeding behavior is not changed in response to the feminization. Measuring rates on infected/sham-infected filters, we found that overall feeding was greatly reduced, compared to normal food, as expected. We did not see a significant difference between feeding rates in males, females or feminized males on either LB or *Ecc*, suggesting that feeding rate is not likely to be a confounding factor leading to greater male sensitivity. N.B. Considering this feeding data with respect to the Smurf assay: lower male feeding rates are not precluding males from showing the Smurf phenotype, as we observed a high number of Smurfs in the feminized males, despite their feeding rates being equal to control males on SYA.

*2) Also, it appears that the assay was carried out at 35d of age. As female flies live longer than males it is difficult to interpret this finding. 35d old females will likely be 'healthier' than 35d old males. At 35 days of age, female flies will likely be more resistant to most extrinsic stressors, including stressors that do not act exclusively on the gut (e.g., heat, hyperoxia). Perhaps 35d old female flies have better immune function than 35d old males?*

We understand the concern that the healthspan of females and males are likely to be different given the differences in lifespan. However, we are wary of performing stress assays on older cohorts that have lost a significant proportion of flies, as what remains can be described as a ‘selected population’ that contains the healthier individuals, the weakest having been lost to early death. We have performed subsequent infections at 42 d, which represents an age where females already show significant gut pathology, but just precedes the steep decline on the male survival curve. We also agree that further analysis of the immune response in aged males and females would be informative, see below.

*3) In Figure 2, it appears that male guts do show an ISC proliferation response to the infection. In fact, it could be argued that the magnitude of the male response is GREATER than that of the female response.*

As a proportion of basal levels, rather than numbers *per se*, male ISCs are indeed more responsive to infection. However, the rate per male gut is usually <10. Although it has been shown that the ability of ISCs to respond to intestinal damage is important for survival (Buchon 2009, Chatterjee 2009), it is not known whether rate of division or total number of stem cells is important (and this may well be context-dependent). We have amended the discussion of this result accordingly.

*In summary, there are a number of reasons why at 35 days of age male flies are more sensitive to this infection. I would suggest that the authors are more careful not to overstate their findings and/or provide stronger evidence to support their claims.*

We strongly agree that sex differences in immune responses may be complex in origin and bear further investigation. We have performed two further experiments as described above: we have tested the resistance to *Ecc* oral infection in males, females and gut-feminized males (described above), and have performed qPCR for systemic *diptericin* (a gram negative responsive, IMD-pathway AMP) induction after *Ecc* oral infection, and *Duox*, which is expressed mainly in the gut epithelium after oral infection by uracil-releasing bacteria such as *Ecc*.

Systemic *diptericin* expression increases after infection in 42 d flies,but is induced at significantly higher levels in males than females. Males also show a high basal level of *diptericin* expression, indicative of systemic inflammation (see analysis inflammation during ageing, in response to Reviewer 3). Taken in context with the sensitivity of aged males to *Ecc* ingestion, high AMP expression may be an indicator of a worse outcome for males after oral infection. Gut feminized males show a high level of *diptericin* induction, in line with their reduced survival. This suggests that feminizing the gut does not feminize other aspects of the response to infection, such as AMP production by the fat body.

*Suggestions to improve impact A potential role for the microbiota in sex differences in gut pathology is ignored. Several studies have reported that the presence of gut-associated microbes contributes to 'gut pathology' in female flies. More specifically, axenic female flies show reduced 'gut pathology' with age (Broderick* et al.*, 2014, Buchon* et al.*, 2009, Guo* et al.*, 2014; Clark* et al. *2015). This is, of course, what the authors claim to observe in the aging intestines of male flies. It would, therefore, be interesting to find out whether male flies show an age-related increase in bacterial load and/or microbial imbalance (dysbiosis) in the strains used in this study. If they do not, this may contribute to the delayed intestinal degeneration observed. I am not suggesting that this needs to be figured out in detail in the current manuscript. But, it is certainly worth investigating and could, potentially, provide insight. It would also be interesting and strengthen the claims if the authors could show that feminized male guts show alterations in the microbiota.*

We have performed a preliminary analysis of microbiota during ageing in male, female and gut feminized flies, as described above in the response to Reviewer 1. We find that males of our *w^Dah^* line have a lower internal bacterial load than do females, and that this trend is reversed in feminized males. This is a very interesting preliminary observation, and we have included it in the manuscript.

However, this rather cursory analysis did not identify strong age-related changes in load (except in gut feminized males), or gross differences in the proportions of aerobic or anaerobic species. That is not to suggest that these features do not exist, and we may well have failed to detect subtle, but important differences between the sexes’ microbiomes. Further study will need to take in to account both behavioral and physiological differences between males and females. Accordingly, we have been cautious in our discussion of these data. We suggest that this is an important and interesting arena for further investigation.

*Reviewer #3: 1) Since the male gut maintains its integrity with age but live shorter the relationship to longevity is not clear, except that the superimposition of a 'feminization' of the gut in a male body seems to make things worse.*

Given the broad body of work showing that gut decline has a direct relationship with female lifespan, and that male guts do not present the same decline, we hypothesize that intestinal dysfunction is not limiting to male lifespan. Of course, something else must be driving male mortality, and this is an open and interesting question. We agree that comparing other measures of decline will be informative, please see below.

*2) Given point 1, other measures of decline should be explored, such as oxidative stress, anti-microbial responsiveness (AMPs, etc.), proteostasis, etc., the authors allude to but do not consider, in order to provide additional direct (or inverse) correlates, other than ISC numbers.*

To assess other measures of decline (and in addition to *diptericin* induction on infection described above) we have measured *Dual oxidase (Duox*), a ROS producer and an indicator of oxidative stress, and systemic *diptericin (dipt*) expression in unchallenged males, females and gut-feminized males at young and old ages to get an idea of inflammatory status. We find that both males and females increase expression of *dipt* with age although, strikingly, males induce *dipt* at a significantly higher level than do females. In addition, males have a higher level of *Duox* expression than do females at young and old ages, likely reflecting intestinal ROS, because the gut is the major site of expression. This points to another intriguing sex bias in immune function, although we do not yet know how this correlates with the observed sex bias in gut decline, particularly given its inverse nature. We suggest sex differences in immunity uncovered here are not solely driven by decline of the gut epithelium given that, despite better barrier function, less gut pathology, lower ISC activity and lower bacterial load, males have high systemic inflammation and ROS and lower resistance to oral infection.

When we analysed systemic *diptericin* during ageing in gut feminized males, we found that expression levels were several fold higher than in control males and several orders of magnitude higher than in females, at both young and old age. This is in line with our findings that microbial load is increased in feminized males compared to males and females and that the gut barrier is compromised earlier than in females. But, as discussed above, we recognize that this may not be a straightforward correlation with gut maintenance, and have therefore been cautious in our discussion.

*Regarding the latter, could the ISC number (or anti-microbial responsiveness) be directly reduced in the female gut or elevated in the male gut, other than (systemic) DR, that may have indirect effects?*

Several studies have manipulated ISC number directly in female guts by gut-specific expression of; a dominant-negative version of the Insulin Receptor, JAK-STAT signalling, and PGC-1, and have increased female lifespan as a result (Biteau 2010, Rera 2011, Ulgherait et al. 2014 and others). We agree that manipulation by another paradigm would strengthen the connection between gut decline and lifespan in both sexes. We have included data from rapamycin treated *Resille-GFP* flies (see response to Reviewer 1), demonstrating a sex bias in the effect on gut pathology/ISC number. Although we recognize that this is also a systemic treatment, the fact that female (and feminized male) gut pathology responds to the treatment supports the idea that rescue of gut maintenance is a driver for extended lifespan in females (and gut-feminized males), but is less important for males.

*3) The high male vs female sensitivity of males to bacterial infection or DDT is interesting but insufficiently explored to be conclusive. Do ISC proliferate more, produce a higher anti-microbial response in females? What happens in young females, are they more susceptible? The discrepancy between susceptibility and gut integrity/permeability is puzzling and in a way counterintuitive, despite the speculation that higher ISC numbers may provide better potential repair (not demonstrated!). What is the progression of gut and other pathologies upon infection or DDT? Does the gut disintegrate more quickly in males with these challenges, or is the resulting morbidity unrelated to the gut?*

We agree that further analysis of this phenotype is important. We have expanded our analysis of the response to oral *Ecc* infection as described in detail above. Females are resistant to *Ecc* infection at young and old ages, induce a high level of ISC proliferation in response to infection (although this is overlaid on a high basal level of activity in old guts), but do not induce a significant systemic AMP response, perhaps suggesting that they have effectively dealt with the infection at the level of the gut. Males increase ISC proliferation, but nevertheless succumb to oral infection at older ages, with a high and variable systemic induction of *diptericin.* Feminized males induce ISC proliferation to a similar level as females, but induce high systemic AMPs like males, and the infection is fatal.

This suggests that the hypothesis we offered, that the observed sex bias is driven by different rates of repair, was over-simplistic, and we have modified the discussion to reflect this. ISC-driven repair has been demonstrated to be required for survival to oral infection in females (Buchon 2009, Chatterjee 2009), however, several other parameters will affect male and female ability to resist infection, including; barrier function at the time of bacterial ingestion, specific bacterial and viral load, basal and induced AMP and *Duox* release and tolerance mechanisms.

*Ecc* ingestion is not usually lethal to healthy female adults (this study, Ha 2005, Regan 2013), and this is true, even at older ages, despite barrier dysfunction in a subset of females demonstrated by the Smurf assay. We agree that this is counter-intuitive, and suggest that a leaky gut barrier is not a driver of mortality in the context of *Ecc* infection in females (although it may well be important in the face of more virulent infections, or in the interaction of the ageing fly with its microbiota).

*4) Controls are missing: Both driver and UAS controls should be provided for all experiments, not just one for longevity studies and the other for other experiments.*

We have added both driver and UAS controls to the manuscript. They were already included in all lifespans (which are particularly sensitive to polygenic effects), and are now presented fully. Both controls have been provided for all other subsequent analyses, except in the cases where the experiment required a large amount of dissection within a small time window, such as ISC analysis after 18h infection. To address this, we have compared ISC activity over ageing in our two control lines, and find no difference between them (this is now presented in the manuscript).

*5) The high-low yeast should be better explained in terms of concentration and how this relates to 1/2SYA in lifespan studies.*

We have amended the text and the figures to make the feeding regime clear. ‘High’ (2SYA) and ‘low’ (1SYA) are equivalent to 1:2 ratio of sugar to yeast and 1:1 sugar to yeast, respectively. An extra sentence has been added to the Methods section to clarify this.

In summary, we feel that the additional data presented here strengthens the study and we thank the reviewers for their comments and suggestions for further work. We have shown that the males and females differ in the pathology of the ageing gut, at least partly driven by sex-biased stem cell activity, and female intestinal decline can be recapitulated in the male gut by tissue-autonomous sex-switching. Lessening of this pathology extends lifespan in females but it may not be a significant driver of male mortality. Feminization of gut cells increases bacterial load, induces gut pathology, shortens male lifespan and induces plasticity to DR, but does not make males more robust to intestinal infection. Aged males succumb to an oral infection that females are resistant to, and induce high levels of inflammatory AMPs and ROS over ageing, raising the intriguing possibility that other sex-biased immune processes are important for male mortality.